# Subjective socioeconomic status moderates depression's impact on fairness perception in the ultimatum game: A moderated mediation model

Yin Hanmo[1,2], Rozainee Khairudin[3], Shi Yixin[4], Zhang Xi[5], Chen Xinyu[1,2], Syakirien Yusoff[1,2,6], Nasrudin Subhi[1,2]*

**1** Centre for Research in Psychology and Human Well-Being (PsiTra), Faculty of Social Science and Humanities (FSSK), Universiti Kebangsaan Malaysia, Bangi, Malaysia, **2** Psychology Program, Faculty of Social Science and Humanities (FSSK), Universiti Kebangsaan Malaysia, Bangi, Malaysia **3** School of Liberal Arts and Sciences, Faculty of Social Sciences and Leisure Management, Taylor's University Malaysia, Subang Jaya, Malaysia, **4** Faculty of Business, Zhengzhou University of Economics and Business, Zhengzhou, China, **5** Hua Shui Primary School of Zheng Dong New District, Zhengzhou, China, **6** Malaysian Armed Forces Headquarters, Kuala Lumpur, Malaysia

* nas2572@ukm.edu.my

## Abstract

Individuals with depression often exhibit cognitive distortions in socioeconomic decision-making, particularly in interpreting fairness. However, the role of subjective socioeconomic status in shaping these distortions remains underexplored. This study investigates how depression influences fairness perception and rejection behavior in the Ultimatum Game. Specifically, it is to examine whether the fairness perception of unfair offers mediates the association between depression and the rejection rate of unfair offers, and whether subjective socioeconomic status moderates this relationship. 274 participants completed the CES-D scale to assess depressive symptoms, the MacArthur Scale to measure subjective socioeconomic status, and participated in a modified UG to evaluate fairness perception and rejection rates. Mediation and moderated mediation analyses were conducted using PROCESS Model 7. The results showed that individuals with higher levels of depression tended to perceive unfair offers as more fair, which subsequently led to fewer rejections. Crucially, this mediation effect was significant only among individuals with high subjective socioeconomic status. For low subjective socioeconomic status individuals, depression did not significantly alter the fairness perception of unfair offers. These findings suggest that subjective socioeconomic status shapes the cognitive consequences of depression, highlighting the importance of accounting for socio-cognitive contextual factors in understanding how depression affects social decision-making processes.

**Data availability statement:** All relevant data are within the manuscript and its Supporting Information files (S1 Data).

**Funding:** The author(s) received no specific funding for this work.

**Competing interests:** The authors have declared that no competing interests exist.

## Introduction

Depression is not only marked by persistent low mood and anhedonia, but also by profound alterations in cognition. A well-documented characteristic of depression is the presence of cognitive distortion, which biases in thinking that negatively affect how individuals perceive themselves, others, and the world [1]. These biases can alter core decision parameters such as perceived value, risk, and fairness. For example, individuals with depressive symptoms may underestimate their own worth, overinterpret threats, or misjudge intentions in social exchanges [2]. The distortion of cognitive systems will greatly affect the individual's social function and lead to a worse social state, which has become an increasingly prominent research question. Specifically, in economic interactions involving fairness, such as the Ultimatum Game (UG), these distortions may lead to atypical behavior, such as accepting or rejecting more unfair offers than common people. The UG is a behavioral experiment in which the proposer proposes an offer to split a sum of money. And the other player, as a responder, chooses to accept the offer and receive the proposed amount or reject it so that both players get nothing [3]. In the UG, it is well replicated in common individuals that low offers (less than 20–30% of the total amount) tend to be rejected by responders [4]. This is thought to relate to participants objecting to 'unfairness' [5], which suggested that individuals willingly incur personal costs to punish unfair or norm-violating behavior, even when they gain less materially from doing so. This action, termed as altruistic punishment [6,7], indicates that the goal isn't profit, but to enforce social norms and fairness. It is suggested that such punitive actions, though seemingly irrational from an economic standpoint, serve a social function: by imposing costs on selfish actors, responders aim to deter future unfairness and promote equitable outcomes in group [8]. Previous research has widely suggested that rejection in the Ultimatum Game may be closely related to emotion [5]. This has been considered a possible pathway to understanding the atypical behaviors caused by depression, as depression significantly distorts the emotional system [9].

Past research has consistently indicated the role of negative emotion in the rejection of unfair offers in the UG. A study by Harlé and Sanfey (2007) found that induced sadness led participants to reject unfair offers more frequently compared to those in neutral emotional states [10]. This suggests that incidental sadness biases social economic decisions in the UG. Similarly, a study by Andrade and Ariely (2009) demonstrated that incidental emotions, such as anger and disgust, increased participants' sensitivity to fairness, leading to higher rejection rates of unfair offers in the UG [11]. Research by Frith & Singer (2008) and Wischniewski et al. (2009) highlights the neural mechanisms behind this behavior [6,7]. Negative emotions activate brain regions associated with conflict resolution (*e.g.*, the anterior insula and prefrontal cortex), creating tension between rational economic decisions (accepting offers for material gain) and emotional impulses to punish inequity. Also, Sanfey et al. (2003) used fMRI approach to show that unfair offers activate the insular, a brain region associated with disgust and emotional arousal [5].

Considering the key role of negative emotions in rejection to unfairness, the inability to properly process negative emotions caused by unfair treatment may be

the cause of depression-related decision-making biases in the UG [5]. Moreover, cognitive models of depression propose a relationship between distorted emotion processing and decision-making, indicating that depressive symptoms involve increased sensitivity to negative aspects of a situation and decreased awareness of positive events and consequences [12], which may explain the higher rejection rate of unfair offers among depressed individuals. These seem to have provided evidence and plausible explanations. However, when we take a closer look at the findings from studies on depression and the UG, the relationship between depression and the rejection of unfair offers appears to be insufficiently clear if viewed solely from an emotional perspective. Indeed, previous studies have shown considerable inconsistency in the findings regarding the effect of depression on the UG behavior. For instance, Harlé et al. (2010) reported that although depressed individuals reported a more negative emotional reaction to unfair offers, they rejected significantly less unfair offers, compared with the controls [13]. And Agay et al. (2008) also found that the rejection rate of depressive patients was significantly lower than that of the healthy controls [14]. While Wang et al. (2014) and Scheele et al. (2013) demonstrated that the rejection rate of the (Major Depressive Disorder) MDD patients was higher than those of the normal controls [15,16]. Destoop et al. (2012) observed no significant difference in the acceptance rates between severe MDD patients and their healthy controls [17]. Therefore, although the emotional defects of depressed individuals are obvious, in the socioeconomic interaction, the behavioral decision-making pattern of depressed individuals is still complex and inconclusive.

It is worth noting that engaging in altruistic punishment relies not only on emotional motivation, but also on perception of fairness [18]. Depression, which is also associated with cognitive distortions [19], may alter the processing of unfair offers, potentially reducing the likelihood of engaging in punishment behavior. While prior research mostly focused on how depression influences self-focused emotion, less is known about how depressive cognition distorts social constructs like fairness perception, which are central to behavioral economic models of cooperation and punishment [4]. There is one study indicated that individuals with depression have lower acceptance rate of unfair proposals in social interaction scenarios because of the weakened association between fairness perception and decision-making behavior [20]. This indicates the influence of depression on fairness perception and the UG decision-making. However, no research has attempted to use fairness perception to establish a link between depression and the rejection of unfair offers. This has prevented us from gaining a deeper understanding of how depression distorts fairness perception and further affects the UG behavior. So, the first aim of the present study is to investigate whether fairness perception serves as a mediator in the relationship between depression and rejection behavior.

A clearer understanding of how depression shapes fairness perception may shed light on the inconsistent findings reported in prior research using the ultimatum game. Notably, a key question is whether there are other potentially important factors in the pathway from depression to fairness perception to rejection behavior. Since it is a socioeconomic decision, individual socioeconomic factors should play a role, which is an issue that has been neglected in past studies on depression and the UG. Socioeconomic status (SES) deserves to be studied because it is a necessary social attribute for anyone in society, and it constantly affects the way and characteristics of interaction between individuals and society [21]. More importantly, SES has a significant impact on individuals' perceived fairness, which is reflected in multiple dimensions such as resource allocation, opportunity acquisition, social support and psychological experience [22]. Therefore, as fairness perception determines the rejection of unfair offers [18], decision making in the UG should also be sensitive to SES. Although many studies have confirmed that SES affects decision-making in the UG [23–25], this aspect has been largely neglected in studies examining depression within the context of ultimatum game tasks. Not considering the SES of individuals in socioeconomic decision-making is a matter of concern, as people inevitably treat social interactions based on their own SES [26]. Thus, the impact of depression on UG behavior in the laboratory will become quite uncontrollable without considering the SES of participants, which might be the core problem of past research.

So, how might SES affect depression's distortion of fairness perceptions? A critical clue is that people in different SES hold different expectations of fairness, which are shaped by their social experiences, perceived control, and exposure to

inequality [22]. Individuals with higher SES tend to expect fair and equitable treatment, likely due to increased autonomy, access to resources, and greater perceived social value [21]. In contrast, individuals with lower SES may develop reduced expectations for fairness, shaped by repeated exposure to social inequality and systemic disadvantage [21,27]. These differing baseline expectations may shape how unfairness is perceived and how individuals respond to unfair treatment. A key point in the psychology of decision-making is that emotional and cognitive responses to unfair treatment are not determined solely by objective conditions, but by the extent to which those conditions deviate from personal expectations [28]. When an individual's experience of unfairness significantly violates their internal standards or assumptions, as is more likely in high-SES individuals who expect fair treatment, the resulting discrepancy may trigger greater cognitive dissonance and emotional distress [26]. Overall, an individual's SES may also influence the processing of unfair offers in the UG, thereby affecting decision-making. Given the profound and prominent nature of this personal attribute in socioeconomic interaction, it should not be overlooked in the UG study.

Therefore, SES and depressive symptoms are likely to synergistically shape fairness perceptions, which in turn affect UG decision-making. It is important to figure out how SES intervenes in the fairness perception distortion caused by depression, which will help us clarify the sources of conflicting findings in previous research on depression and decision-making in the UG. Specifically, it is worth examining whether different levels of SES may either amplify the perception of unfairness (compensatory effect) or attenuate it (diminishing effect) among individuals with depression. Therefore, the second aim of the present study is to investigate how SES moderates the relationship between depression and fairness perception.

This study examined university students with subclinical depression, a prevalent condition in higher education contexts. Such depressive symptoms may considerably affect academic achievement and everyday functioning, thereby posing long-term risks to students' educational and career trajectories. The Center for Epidemiologic Studies Depression Scale (CES-D) is widely recognized for its effectiveness in identifying both clinical and subclinical levels of depression [29]. Furthermore, it allows it to capture a broad spectrum of depressive symptoms, making it particularly suitable for identifying subclinical depression in research settings. Therefore, this study used CES-D to measure the participants' depression level. Moreover, Adler et al. (2000) argued that subjective SES reflects an individual's internalized perception of social standing and has stronger predictive power for psychological and health-related outcomes than objective SES [30]. Similarly, Singh-Manoux et al. (2005) demonstrated that subjective SES is a better predictor of mental health indicators, particularly in younger populations such as university students [31]. Given the transitional nature of university students' socioeconomic circumstances, subjective SES may serve as a more accurate and psychologically meaningful indicator compared to objective measures [30,31]. So, as university students do not have a fixed income or occupation, and have similar educational levels, this study employed the MacArthur Scale of Subjective Social Status [30] to assess participants' SES. And to enable large-scale data collection and model construction, this study employed a scale-based version of the Ultimatum Game, following the paradigm established by a previous study, which has been shown to be effective in measuring participants' UG behavior tendencies under careful experimental settings [32]. Given that research on the UG has primarily focused on individuals' responses to unfair offers [3,5,8], the present study adopts fairness perception and rejection rates of unfair offers as key variables.

Overall, past studies have shown that depression has a certain degree of impact on UG decision-making, and fairness perception plays an important role in it [16,20]. However, the understanding of this mechanism is incomplete. Because subjective SES is a salient characteristic of each individual and has a strong influence on socioeconomic decision-making [21], ignoring the role of subjective SES will make any explanation of socioeconomic interactions not grounded enough. Therefore, the present study intended to further examine the relationship between depression, fairness perception, UG decision-making and subjective SES. It was hypothesized that fairness perception would mediate the relationship between depression and rejection rate of unfair offers, and that subjective SES would moderate the indirect association between depression and the rejection of unfair offers via fairness perception. This moderated mediation model is displayed in Fig 1.

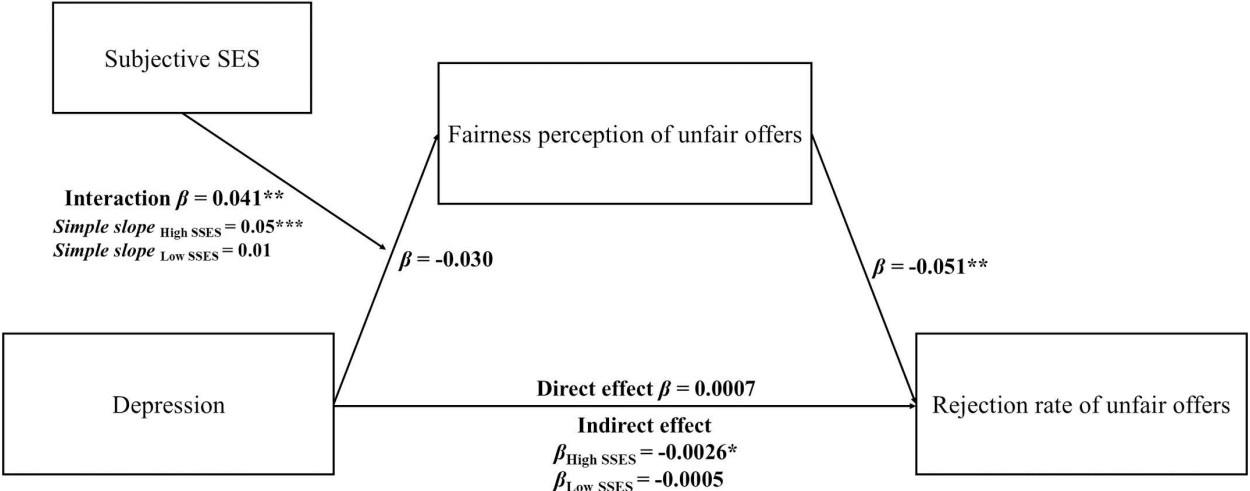

**Fig 1. The conceptual framework of the moderated mediation model.** The path coefficients are standardized beta weights. * $p < 0.05$, ** $p < 0.01$, *** $p < 0.001$.

## Materials and methods

### Participants

University students from The National University of Malaysia and Zhengzhou University of Economics and Business were recruited as participants in this study. Participants were Chinese students and Malaysian ethnic Chinese students, all of whom were native Chinese speakers and could fully understand the questionnaire. A total of 300 students participated in this study. But 26 participants were excluded because they selected the same option for most questions, representing an 91.33% effective response rate (N = 274, Mean age = 20.86, SD age = 3.30, 173 Males). Data collection took place from December 2024 to March 2025. All participants provided written informed consent. This study was approved by Research Ethics Committee, The National University of Malaysia (Protocol Number: JEP-2024–296). All participants who completed the study received the corresponding amount of money according to their choices in the UG.

### Measures

**Depression.** The Center for Epidemiologic Studies Depression Scale (CES-D) scale was developed to screen for depression by measuring the frequency of events and ideas over the past week (Radloff, 1977). The CES-D scale is a 20-item instrument with each item rated on a four-point scale ranging from 0 ("rarely or none of the time") to 3 ("most or all of the time"). Four of the items are positive statements which are inversely scored for calculating the total score. The total score ranges from 0 to 60 and a higher score indicates a greater risk of depression. A total score of 16 or greater is considered as indicative of subthreshold depression. This study used the Chinese version of CES-D, which has been validated in a variety of Chinese samples [33,34]. The Cronbach's alpha coefficient for CES-D in this study was 0.924.

**Fairness perception.** The measurement of fairness perception draws on the past study [20]. Participants were asked to quantify their perceived fairness of all 9 offers presented in the UG task by a 7-point Likert scale ranging from 1 (very unfair) to 7 (very fair). Each offer represented a proposed division of an initial endowment of 100 cents (1 Chinese Yuan), with varying amounts allocated to the responder (e.g., 10:90, 20:80, 30:70, 40:60, 50:50, 60:40, 70:30, 80:20, 90:10).

**Subjective SES.** The traditional 10-rung MacArthur scale [30] was used to assess participants' subjective SES. They were asked to choose the rung that best represented their position in the social hierarchy relative to others in society.

Higher numbers indicated a higher social standing. To determine the threshold for the high and low groups, we calculated the mean and median of subjective SES, which were 5.07 and 5, respectively. Therefore, we set 5 as the cut-off point. Participants with scores above five were in the high subjective SES group (N = 108, Mean = 7.11, SD = 1.35), and those with scores below five were in the low subjective SES group (N = 166, Mean = 3.75, SD = 1.24). Independent sample t test showed significant difference between the two groups ($t(272) = -21.20$, $p < 0.001$).

**UG task and procedure.** To obtain a larger sample of data for modeling, the UG task used a carefully organized paper questionnaire modified from [32]. To enhance the questionnaire's validity, an experienced data collector guided and supervised participants in completing the paper-based survey. Participants were recruited via online chat groups. And they took part in the study in groups of 10–30 individuals, scheduled based on their availability. The experiment took place in a private discussion room in the library. A total of 18 sessions were conducted, each lasting approximately 10 minutes. The average payment per participant was 5.89 Chinese Yuan (SD = 2.73).

Participants were invited to a separate quiet room and introduced to the ultimatum game by the experimenter. Participants were presented with a table with various offers from randomly matched anonymous proposers. In the questionnaire paradigm, the participant decided whether to accept or reject each of the nine possible offers (10:90, 20:80, 30:70, 40:60, 50:50, 60:40, 70:30, 80:20, 90:10). And offers in which the proposer receives more than the responder are considered unfair offers. Each offer was offered twice, so there were 18 proposals in total. The order of offer was randomized. Participants were told that these offers were collected from 18 anonymous proposers in the previous study, and they could accept or reject the offers. If they accepted an offer, they and the proposers would get money according to the proportion of the offers. If they rejected it, neither they nor the proposers would get any money. Participants then indicate whether they reject or accept the offer by marking an '×' or a '√' on the sheet (Table 1). The final payment was calculated based on the sum of all accepted offers.

After completing the ultimatum game, they were asked to complete the Fairness Perception Scale, the CES-D Scale, the MacArthur scale, and a demographic questionnaire (S1 Appendix).

## Data analysis

Descriptive statistics and a correlation analysis of the data were performed using SPSS 24.0., and PROCESS v4.2 was used for mediating and moderating effect analysis [35]. Fairness perception was examined as a mediator between depression and the rejection rate of unfair offers. Gender and age were entered as covariates.

To evaluate the robustness of the estimated effects to potential omitted variable bias, a post-hoc sensitivity analysis based on the approach of Cinelli and Hazlett (2020) was conducted using the R package *sensemakr* [36]. This analysis quantifies how strongly an unmeasured confounder would need to be associated with both the treatment and outcome to

**Table 1. Ultimatum Game form.**

| Proposer No. | Proposer gets (cents) | You get (cents) | If you think it is OK, please tick √. If not, please tick × | Proposer No. | Proposer gets (cents) | You get (cents) | If you think it is OK, please tick √. If not, please tick × |
|---|---|---|---|---|---|---|---|
| 936 | 50 | 50 | | 633 | 90 | 10 | |
| 357 | 70 | 30 | | 377 | 70 | 30 | |
| 118 | 90 | 10 | | 314 | 10 | 90 | |
| 330 | 30 | 70 | | 256 | 60 | 40 | |
| 267 | 10 | 90 | | 127 | 80 | 20 | |
| 630 | 30 | 70 | | 904 | 50 | 50 | |
| 109 | 20 | 80 | | 812 | 40 | 60 | |
| 531 | 80 | 20 | | 513 | 40 | 60 | |
| 221 | 60 | 40 | | 101 | 20 | 80 | |

invalidate the causal conclusions. Partial R² and robustness value (RV) were reported to assess the vulnerability of the effect, higher RV indicate greater robustness to omitted variable bias. Our analysis focused on two key pathways within the moderated mediation framework [37]: (1) the moderated effect of depression by subjective SES on the mediator, fairness perception of unfair offers, and (2) the subsequent effect of unfairness perception on the outcome variable, rejection rate of unfair offers. Sensitivity analyses were conducted to assess the robustness of these pathways to potential unobserved confounding.

## Results

### Descriptive statistics

The descriptive statistics and correlations of each variable are shown in Table 2. And rejection rates as well as fairness perceptions for each of the nine possible offers are shown in Table 3. Depression was significantly positively correlated with gender ($r=0.12$, $p<0.05$). The fairness perception of unfair offers was significantly positively correlated with gender ($r=0.13$, $p<0.05$), depression ($r=0.28$, $p<0.01$) and subjective SES ($r=0.13$, $p<0.05$). The rejection rate of unfair offers was significantly negatively correlated with the fairness perception of unfair offers ($r=-0.20$, $p<0.01$).

### Mediation test of fairness perception of unfair offers

The mediating effect of fairness perception was analyzed while controlling for gender and age. We calculated 95% confident intervals (CI) based on a 5000 bootstrap resampling. The data were processed using PROCESS model 4, and the results showed (Table 4) that depression had a significant positive effect on the fairness perception of unfair offers ($\beta=0.031$, $p<0.001$). And the fairness perception of unfair offers had a significant negative effect on the rejection rate

**Table 2. Descriptive statistics and correlation analysis.**

| Variables | 1 | 2 | 3 | 4 | 5 | 6 | M | SD |
|---|---|---|---|---|---|---|---|---|
| 1. Age | 1 | | | | | | 20.86 | 3.3 |
| 2. Gender | −0.09 | 1 | | | | | 1.37 | 0.48 |
| 3. Depression | −0.11 | 0.12* | 1 | | | | 18.11 | 11.75 |
| 4. Subjective SES | 0.11 | −0.06 | 0.05 | 1 | | | 1.39 | 0.49 |
| 5. Fairness perception of unfair offers | −0.05 | 0.13* | 0.28** | 0.13* | 1 | | 2.71 | 1.39 |
| 6. Rejection rate of unfair offers | −0.05 | −0.07 | −0.03 | −0.02 | −0.20** | 1 | 0.58 | 0.36 |

Note. Male was counted as 1, female as 2; low subjective SES was counted as 1, high subjective SES was counted as 2. * $p<0.05$, ** $p<0.01$.

**Table 3. Rejection rates and fairness perceptions for each of the nine possible offers.**

| Offer (Proposer:You) | Rejection rate | | Fairness perception | |
|---|---|---|---|---|
| | *M* | *SD* | *M* | *SD* |
| 10:90 | 0.40 | 0.21 | 3.06 | 1.99 |
| 20:80 | 0.36 | 0.19 | 3.25 | 1.80 |
| 30:70 | 0.27 | 0.15 | 3.49 | 1.71 |
| 40:60 | 0.22 | 0.15 | 4.00 | 1.60 |
| 50:50 | 0.14 | 0.09 | 5.61 | 1.68 |
| 60:40 | 0.34 | 0.21 | 3.30 | 1.39 |
| 70:30 | 0.57 | 0.21 | 2.85 | 1.57 |
| 80:20 | 0.68 | 0.19 | 2.51 | 1.55 |
| 90:10 | 0.75 | 0.16 | 2.18 | 1.61 |

**Table 4. Mediating effect of the fairness perception of unfair offers.**

| Variables | | Overall fit index | | | 95%CI | | | |
|---|---|---|---|---|---|---|---|---|
| Result | Predictor | R | R² | F | β | LLCI | ULCI | t |
| Fairness perception of unfair offers | Depression | 0.293 | 0.086 | 8.470*** | 0.031 | 0.017 | 0.045 | 4.479*** |
| | Age | | | | −0.007 | −0.056 | 0.042 | −0.285 |
| | Gender | | | | 0.272 | −0.06 | 0.604 | 1.613 |
| Rejection rate of unfair offers | Depression | 0.211 | 0.045 | 3.146* | 0.001 | −0.003 | 0.004 | 0.342 |
| | Fairness perception of unfair offers | | | | −0.051 | −0.083 | −0.02 | −3.167** |
| | Age | | | | −0.006 | −0.019 | 0.007 | −0.97 |
| | Gender | | | | −0.042 | −0.131 | 0.047 | −0.934 |
| Rejection rate of unfair offers | Depression | 0.095 | 0.009 | 0.824 | −0.001 | −0.005 | 0.003 | −0.501 |
| | Age | | | | −0.006 | −0.019 | 0.007 | −0.901 |
| | Gender | | | | −0.056 | −0.146 | 0.034 | −1.229 |

Note. β = standardized regression coefficient; LLCI = lower limit of the 95% confidence interval; ULCI = upper limit of the 95% confidence interval; * $p < 0.05$, ** $p < 0.01$, ***$p < 0.001$.

of unfair offers (β = −0.051, p < 0.01). Although the indirect effect of depression on the rejection rate through the fairness perception of unfair offers was small in magnitude, it was statistically significant as the 95% bootstrap confidence interval did not include zero (β = −0.0016, 95% CI = [−0.0032, −0.0004]), suggesting a subtle but reliable mediation effect [35,38]. And the direct effect of depression on the rejection rate of unfair offers was not significant (β = 0.0007, 95% CI = [−0.0031, 0.0044]), suggesting the presence of full mediation. This result indicates that depression may influence rejection behavior fully through the fairness perception of unfair offers.

### Test of moderated mediation effect

To prevent multi-collinearity issues, all continuous independent variables were mean-centered before creating the interaction terms [39]. The data were processed using Process model 7 to test whether subjective SES moderated the relationship between depression, fairness perception of unfair offers, and the rejection rate of unfair offers. The results (Table 5) suggested that subjective SES positively predicted the fairness perception of unfair offers significantly (β = 0.358, p < 0.05) and the fairness perception of unfair offers negatively predicted the rejection rate of unfair offers significantly (β = −0.051, p < 0.01). The product term of depression and subjective SES positively predicted the fairness perception of unfair offers significantly (β = 0.041, p < 0.01), demonstrating that subjective SES has a moderating effect on the impact of depression on the fairness perception of unfair offers. Simple slope analyses were utilized to demonstrate significant interaction at high and low subjective SES (Fig 2). The results showed that, for individuals with high subjective SES, higher depression a was associated with higher fairness perception of unfair offers (simple slope = 0.05, t = 5.24, p < 0.001). However, no significant relationship between depression and the fairness perception of unfair offers was found (simple slope = 0.01, t = 1.11, p = 0.27) when subjective SES was low. Conditional indirect effect analysis revealed that among individuals with high SES, depression indirectly decreased the rejection rate of unfair offers via fairness perception (indirect effect = −0.0026, 95% CI [−0.0049, −0.0007]), whereas this effect was nonsignificant among low SES individuals (indirect effect = −0.0005, 95% CI [−0.0020, 0.0007]).

### Sensitivity analysis

The statistical results are shown in Table 6 and Fig 3. The moderation effect of Depression (X) × Subjective SES (W) on the fairness perception of unfair offers (M) demonstrated moderate robustness to omitted variable bias, with a partial R² of 3.31% and a robustness value (RV) of 16.87% needed to reduce the effect to zero. Similarly, the path from the fairness

**Table 5. Mediated model tests with moderation.**

| Variables | | Overall fit index | | | 95%CI | | | |
|---|---|---|---|---|---|---|---|---|
| Result | Predictor | R | R² | F | β | LLCI | ULCI | t |
| Fairness perception of unfair offers | Depression | 0.363 | 0.132 | 8.155*** | −0.03 | −0.072 | 0.011 | −1.442 |
| | SSES | | | | 0.358 | 0.038 | 0.679 | 2.199* |
| | Depression*SSES | | | | 0.041 | 0.014 | 0.067 | 3.028** |
| | Age | | | | −0.014 | −0.062 | 0.034 | −0.57 |
| | Gender | | | | 0.268 | −0.058 | 0.593 | 1.619 |
| Rejection rate of unfair offers | Depression | 0.211 | 0.045 | 3.146* | 0.001 | −0.003 | 0.004 | 0.342 |
| | Fairness perception of unfair offers | | | | −0.051 | −0.083 | −0.02 | −3.167** |
| | Age | | | | −0.006 | −0.019 | 0.007 | −0.97 |
| | Gender | | | | −0.042 | −0.131 | 0.047 | −0.934 |

Note. β = standardized regression coefficient; LLCI = lower limit of the 95% confidence interval; ULCI = upper limit of the 95% confidence interval; * $p < 0.05$, ** $p < 0.01$, ***$p < 0.001$.

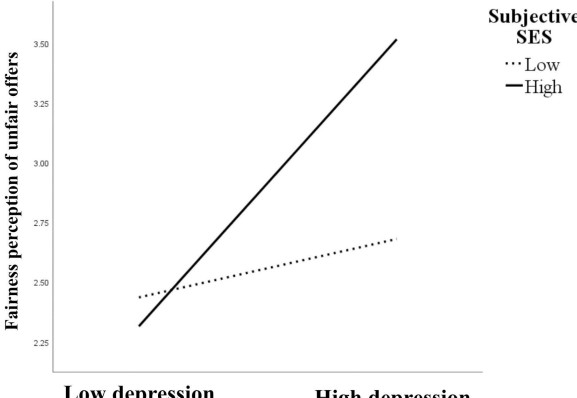

**Fig 2. Subjective SES moderates the effect of depression on the fairness perception of unfair offers.**

**Table 6. Sensitivity analysis results for the model.**

| Pathway | Coefficient Estimate | Partial R² (%) | Robustness Value to 0 (%) | Robustness Value to Insignificance (%) |
|---|---|---|---|---|
| X×W→M | 0.04 | 3.31 | 16.87 | 6.24 |
| M→Y | −0.05 | 3.58 | 17.51 | 6.98 |

perception of unfair offers (M) to the rejection rate of unfair offers (Y) showed a robust effect with a partial R² of 3.58% and an RV of 17.51%. We observed that all robustness values (RVs) exceeded the partial R² values of key theoretical predictors in our model. Overall, these findings suggested that the moderated mediation model was reasonably robust against unmeasured confounding, supporting the validity of the observed effects.

## Discussion

This study formulated a moderated mediation model to investigate the influence of depression, subjective SES and fairness perception on the rejection behavior in the UG. We found the mediating effect of the fairness perception of unfair

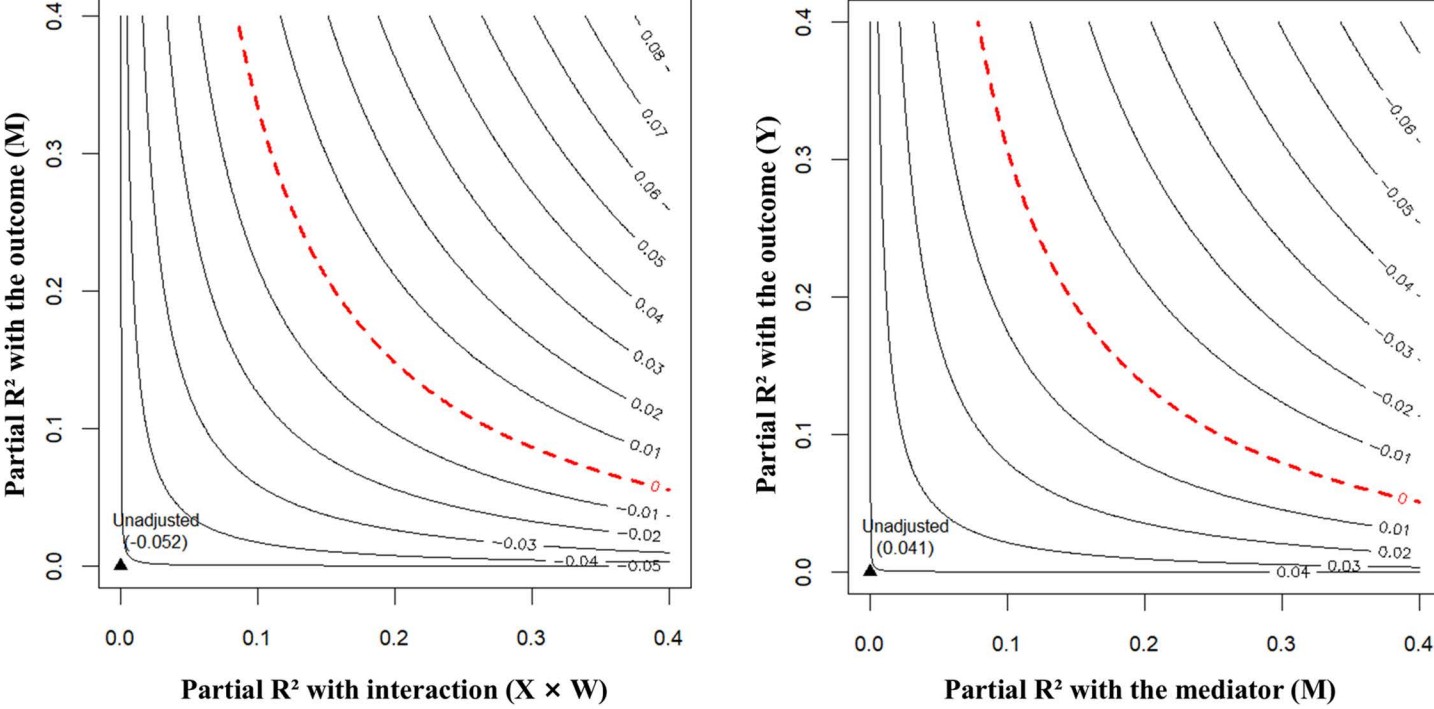

**Fig 3. Sensitivity contour plots.** The x-axis shows the partial R² of an unobserved confounder with the treatment; the y-axis shows its partial R² with the outcome. The contour lines denote the strength needed to reduce the estimate to zero (dashed) or to statistical insignificance (solid).

offers on the association between depression and the rejection rate of unfair offers, and the moderating effect of subjective SES on the path between depression and the fairness perception of unfair offers. These results further deepen the understanding of the relationship between depression and socioeconomic decision-making, especially highlighting the interactive mechanism between emotional and social factors.

As expected, our study found that the impact of depression on the rejection rate of unfair offers was mediated by the fairness perception of unfair offers. While the indirect effect was significant, the direct effect of depression on the rejection rate of unfair offers did not reach statistical significance. This result suggests that the direct link between depression and the rejection of unfair offers is relatively limited. As previous research has found, although depression can induce stronger negative emotions toward unfair offers, it did not directly lead to more rejection tendencies [13]. These findings further elucidate the cognitive-behavioral mechanisms underlying the relationship between depression and the rejection to unfair offers, suggesting that such rejection is not merely driven by negative emotional responses but involves more nuanced cognitive evaluations [40].

Therefore, the effect of depression on rejection to unfairness appears to depend on the presence of distorted perceived unfairness. Specifically, individuals with higher levels of depression were more likely to perceive unfair offers as fair (increased fairness perception), which in turn led to a lower likelihood of rejecting such unfair offers. To a certain degree, the result aligns with earlier research by Harlé et al. (2010) and Agay et al. (2008), which reported lower rejection rate of unfair offer among individuals exhibiting elevated depressive symptoms [13,14]. Importantly, both studies recruited subclinical samples, as in the present study, suggesting that this effect may be a characteristic specific to this population. Harlé et al. (2010) interpreted the phenomenon that depressed individuals exhibit stronger negative emotional responses to unfairness but show lower rejection rates as a result of depression-related emotion regulation and rationalization

mechanisms [13]. Specifically, individuals with depression may actively rationalize unfair offers as more acceptable in order to reduce their emotional conflict and avoid internal distress. They may employ cognitive emotion regulation strategies, such as reappraisal, to mitigate their emotional responses by reframing the situation as "It's not that unfair after all." Such an increased perceived fairness of unfair offers related to depression has also been documented in a recent study conducted with a university student subclinical sample [41], which further highlights the specific association between depression and the fairness perception of unfair offers. This finding contributes to a growing body of literature indicating that depression is associated with cognitive distortions in social judgment [42], which highlights a nuanced interplay between emotion and cognition in shaping socioeconomic decision-making.

However, studies involving clinically diagnosed MDD samples have found opposite findings. As reported by Wang et al. (2014), Jin et al. (2022) and Scheele et al. (2013), individuals with MDD exhibited higher rejection rates in response to unfair offers [15,16,20]. This implies that the severity of depression may exhibit a non-linear relationship with the tendency to reject unfair offers. We speculate that there may be a complex process involved in the effect of depression on regulatory function. In cases of minor depression like present study [13,14], regulatory processes may be distorted into maladaptive defense mechanisms, thereby reducing the likelihood of rejecting unfair offers relative to non-depressed individuals. However, as depressive severity increases, these regulatory functions become substantially impaired [43], ultimately resulting in heightened rejection rates of unfair offers. Given that past studies have used different criteria for identifying depression and the range of participants recruited, the severity of depression should also be a factor to consider in order to obtain a reasonable and reliable argument.

More importantly, the results revealed a significant interaction effect between depression and subjective SES on the fairness perception of unfair offers. Consequently, the mediating effect of the fairness perception of unfair offers in the relationship between depression and the rejection rate of unfair offers was observed exclusively among individuals with high subjective SES. Specifically, only among participants with high subjective SES did depressive symptoms lead to increased fairness perception of unfair offers, which in turn resulted in lower rejection rate. Conversely, among individuals with low subjective SES, this pathway did not operate, leading to a lack of association between depressive symptoms and the rejection rate of unfair offers. We propose that the divergence in past research on depression and rejection of unfairness may come from this potential interactive mechanism. The lack of an observed association between depression and the rejection rate of unfair offers in some previous studies may be due to the confounding effect of participants' subjective SES. In particular, when participants' overall subjective SES is not sufficiently high, depressive symptoms may have little impact on the fairness perception of unfair offers and thus may not translate into differences in rejection behavior. Future research could further examine this causal relationship by manipulating subjective SES.

A possible explanation is that individuals with different subjective SES have different expectations of unfair treatment. High SES individuals typically expect equitable outcomes and respect, while low SES individuals often anticipate inequality or discrimination, shaped by chronic exposure to social disadvantage [21,27]. This leads to different levels of expectation violation when facing the same unfair treatment. If there are no depressive symptoms, then the individual can handle and process this cognitive conflict well. However, when experiencing depression, individuals with higher subjective SES may employ cognitive reappraisal strategies in a maladaptive form, by reinterpreting unfair treatment as fair, to reduce cognitive dissonance [44,45]. In contrast, individuals with lower subjective SES, who may be more frequently exposed to structural inequities, might show greater tolerance for unfairness. As their degree of cognitive conflict is less, they do not require a lot of cognitive adjustment. Therefore, depression's distortion of fairness perception is less likely to emerge in low-SES populations.

Taken together, we propose that cognitive reappraisal may be a key factor underlying the distortion of rejection behavior toward unfair offers in individuals with subclinical depression. This cognitive reappraisal primarily functions by reframing unfairness as fair, in order to alleviate the negative emotions and internal conflict elicited by unfair treatment. Although it functions as a defense mechanism as it quickly relieves discomfort [45], it does have an impact on actual behavior,

causing decision-making deviations from the non-depressed population. Notably, there is an important threshold for the emergence of this kind of cognitive regulation caused by depression. That is, it requires a higher fairness expectation conflict, which is mainly reflected in the group with higher subjective SES. This mechanism makes it possible for individuals with high subjective SES to be unable to make correct judgments about socioeconomic interactions when they suffer from depression, which in turn leads to the inability to implement reasonable altruistic punishment. Past studies have focused on the impairment of emotional functions caused by depression, arguing that depression may cause individuals to have stronger negative emotions in response to unfair treatment [46], thereby making individuals oversensitive to unfairness [15]. Such perspectives neglect the possibility that individuals with varying subjective SES may appraise unfair situations through different lenses of perceived fairness, shaped by their social standing and lived experiences.

In addition, this study yielded a noteworthy finding: no significant association was observed between depression and subjective SES ($r = 0.05$, $p > 0.05$), which contrasts with previous literature that has typically reported a negative correlation between depressive symptoms and subjective SES [30,31]. This may be because the status of university students' subjective SES in their psychological structure is peculiar, resulting in less correlation with emotional status. Depression among university students is often influenced by short-term environmental factors such as academic pressure and social anxiety [47,48], while the impact of long-term factors like subjective SES tends to be relatively weaker. Moreover, although these students come from diverse family backgrounds, they attend the same university, which indicates a similar level of education. This also makes subjective SES a buffer against the distortion of depression. Rather than being an isolated case [49], this phenomenon reflects a broader pattern that warrants deeper exploration of its underlying mechanisms in future studies.

This study also has certain shortcomings. First, this study employed a self-report screening instrument designed to assess subclinical levels of depression, which may limit the generalizability of the findings to clinically diagnosed populations. Future studies can include a group that has been diagnosed with MDD for comparative analysis. Second, given that the participants were university students, this study focused on subjective SES. Future research should include more diverse populations and investigate the effects of objective social class or other relevant attributes. In addition, based on the results, this study proposes a theoretical hypothesis that individuals with high subjective SES have higher expectations of fairness, which leads to greater perceived fairness conflicts. However, no direct measurement was conducted, and future research could validate this hypothesis through behavioral experiments.

## Conclusion

In conclusion, the present study investigated the underlying mechanisms of the association between depression and the decision-making in the UG and constructed a structural model of depression, the fairness perception of unfair offers, subjective SES, and the rejection rate of unfair offers. This study found that depression has an indirect effect on the rejection rate of unfair offers through fairness perception. And subjective SES plays a moderating role in the predictive effect of depression on the fairness perception of unfair offers. To be exact, this study demonstrates that depression can distort and inflate individuals' perception of unfairness, making unfair offers appear more acceptable. Notably, such distortion was observed exclusively in individuals with high subjective SES. Future research needs to consider individual socioeconomic background factors to explore whether these factors affect individuals' prior expectations of fairness and whether differences in expectations interfere with decision making.

## Supporting information

**S1 Appendix. Experimental questionnaires.**
(DOCX)

**S1 Data. The original dataset.**
(XLSX)

## Author contributions

**Conceptualization:** Yin Hanmo, Rozainee Khairudin, Nasrudin Subhi.

**Data curation:** Yin Hanmo, Rozainee Khairudin.

**Formal analysis:** Yin Hanmo.

**Investigation:** Yin Hanmo, Rozainee Khairudin, Shi Yixin, Zhang Xi, Chen Xinyu, Syakirien Yusoff.

**Methodology:** Yin Hanmo.

**Project administration:** Rozainee Khairudin, Nasrudin Subhi.

**Resources:** Nasrudin Subhi.

**Software:** Yin Hanmo.

**Supervision:** Rozainee Khairudin, Nasrudin Subhi.

**Validation:** Rozainee Khairudin, Nasrudin Subhi.

**Visualization:** Yin Hanmo.

**Writing – original draft:** Yin Hanmo.

**Writing – review & editing:** Rozainee Khairudin, Nasrudin Subhi.

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
