## [Decision Letter · Decision Letter 0]

26 Jun 2025

PONE-D-25-24754Subjective socioeconomic status moderates depression's impact on fairness perception in the ultimatum game: a moderated mediation modelPLOS ONE

Dear Dr. Subhi,

Thank you for submitting your manuscript to PLOS ONE. After careful consideration, we feel that it has merit but does not fully meet PLOS ONE’s publication criteria as it currently stands. Therefore, we invite you to submit a revised version of the manuscript that addresses the points raised during the review process. I have received reports from 2 expert reviewers. Both reviewers recommend a revision. The reviewers agree the research question is worthwhile, and the paper makes a useful contribution to the literature. However, several aspects need clarification, including the experimental procedures, statistical analysis, and discussion of the conclusions, as detailed in the reports. In your revisions, please include a response to each of the reviewers' comments. 

We look forward to receiving your revised manuscript.

Kind regards,

Caleb Cox

Academic Editor

PLOS ONE

Journal Requirements:

Reviewers' comments:

Reviewer's Responses to Questions

**Comments to the Author**

1. Is the manuscript technically sound, and do the data support the conclusions?

Reviewer #1: Yes

Reviewer #2: Partly

2. Has the statistical analysis been performed appropriately and rigorously? 

Reviewer #1: Yes

Reviewer #2: Yes

3. Have the authors made all data underlying the findings in their manuscript fully available?

Reviewer #1: Yes

Reviewer #2: Yes

4. Is the manuscript presented in an intelligible fashion and written in standard English?

Reviewer #1: Yes

Reviewer #2: No

5. Review Comments to the Author

Reviewer #1: This study investigates the effect of depression on rejection behavior in the Ultimatum Game (UG), focusing on how fairness perception mediates responses to unfair offers. It also examines subjective Socioeconomic Status (SES) as a moderator in the relationship between depression, fairness perception, and rejection behavior. The authors find that higher levels of depression are associated with perceiving unfair offers as more fair, which in turn leads to lower rejection rates. The paper contributes meaningfully to the literature. However, several aspects of the experimental design and reporting require clarification and additional information.

Major Comments

1. The authors should report detailed information about the fairness perception task, such as all offer amounts and the initial endowment used in the task.

2. More detailed information is needed about the basic experimental setup. Please include the number of sessions, average duration of each session, average participant payment, and average rejection rates for each of the nine possible offers in the UG task.

3. The incentive structure is unclear. Please explain in more detail how participants were compensated. Was the payment based on one randomly selected decision from the 18 tasks, or were all decisions used to calculate the final payment?

4. The rationale for using a threshold of 5 to define high SES is unclear. Other criteria such as the mean or median SES score could also be considered. The authors should report the median SES value and clarify whether the results are robust to alternative thresholds (e.g., using the mean or median). Would the main findings hold under a different SES cutoff?

Minor Comments

1. Does the mention of statistical significance in line 252 refer to the 10% level? Please clarify.

2. Reference 30 appears to be missing from the list.

3. The reference list should be ordered numerically.

Reviewer #2: This study examined the moderating effect of subjective SES on the relationship between depression and perceptions of fairness. In turn, perceptions of fairness influenced the decision to accept or reject offers in the ultimatum game. The moderation effect is interesting and demonstrates ways that personal experiences can affect how depression influences cognitive distortions. However, I have some remaining questions about the interpretation as well as some recommendations to improve the clarity of the manuscript.

1. I was expecting that depression would be a predictor of UG rejection rates, although in your introduction you wrote that there is mixed evidence of this relationship. This result has some implications for the hypothesis tested, but it is not directly stated in the results or discussed in context of the previous literature. I would like you to add a bit more in the results and discussion regarding the lack of this effect and why that might be.

2. Is there any reason to believe that depression would affect subjective SES? Given that we expect some distortion about the perceptions of the world in depression, could that alter the subjective experience of SES? Could you provide more information about the relationship between the two?

3. I’d like if the model could include the standardized beta weights in the figure 1 to make it easier to interpret. While it’s not necessary, you may also want to consider other tools to build the model such as powerpoint or google slides to make that easier.

4. Overall, the article could use some additional editing for grammar and clarity. There are several incomplete sentences (ex. “And fairness perception of unfair offers had a negative effect on rejection rate on unfair offers (β = -0.051, p < 0.01).”) that make it a bit harder to read. The writing in the introduction and discussion could also be more concise to get the point across as well. Right now, I had to read it several times to make sure I understood the argument being made. I believe you are arguing that the inconsistent relationship between depression and acceptance of unfair offers is due to differences in perceptions of fairness affected by the interaction of life experiences and cognitive distortions in depression, rather than solely depression. However, there were some other parts of the discussion that I had trouble fitting into the argument, such as the influence of medication and defense mechanisms. Please review these sections to ensure that they are concise and clear about the interpretation of your results.

5. There was a specific sentence that I found difficult to parse: “Depression is a defense mechanism for fairness perception”. I think this might be reversed from your meaning, although I’m not entirely sure. From the other points you’ve made, I would have thought that cognitive distortions to view the world as more fair would be the defense mechanism for depression. Could you please clarify?

6. Please provide a sensitivity analysis for the moderated mediation based on your sample size.

6. PLOS authors have the option to publish the peer review history of their article (what does this mean? ). If published, this will include your full peer review and any attached files.

**Do you want your identity to be public for this peer review?** For information about this choice, including consent withdrawal, please see our Privacy Policy .

Reviewer #1: No

Reviewer #2: No

---

## [Author Response · Author response to Decision Letter 1]

27 Jul 2025

Editor comments:

Response: we have carefully reviewed the PLOS ONE style templates and ensured that our manuscript now complies with the formatting guidelines.

2. Please include captions for your Supporting Information files at the end of your manuscript, and update any in-text citations to match accordingly.

Response: We have added captions for all Supporting Information files at the end of the manuscript and updated the in-text citations accordingly.

Reviewer 1 comments:

Major Comments:

1. The authors should report detailed information about the fairness perception task, such as all offer amounts and the initial endowment used in the task.

Response: We express our gratitude to you for acknowledging this observation. We have added a more detailed description of the fairness perception task in the revised manuscript (lines 243–247): “Participants were asked to quantify their perceived fairness of all 9 offers presented in the UG task by a 7-point Likert scale ranging from 1 (very unfair) to 7 (very fair). Each offer represented a proposed division of an initial endowment of 100 cents (1 Chinese Yuan), with varying amounts allocated to the responder (e.g., 10:90, 20:80, 30:70, 40:60, 50:50, 60:40, 70:30, 80:20, 90:10).”.

2. More detailed information is needed about the basic experimental setup. Please include the number of sessions, average duration of each session, average participant payment, and average rejection rates for each of the nine possible offers in the UG task.

Response: We greatly appreciate your valuable suggestion. We have added experimental sessions and participation payment details (lines 261–265): “Participants were recruited via online chat groups. And they took part in the study in groups of 10 to 30 individuals, scheduled based on their availability. The experiment took place in a private discussion room in the library. A total of 18 sessions were conducted, each lasting approximately 10 minutes. The average payment per participant was 5.89 Chinese Yuan (SD = 2.73).”.

And a new table has been added to show the average rejection rate and the fairness perception of each offer (Table 3).

3. The incentive structure is unclear. Please explain in more detail how participants were compensated. Was the payment based on one randomly selected decision from the 18 tasks, or were all decisions used to calculate the final payment?

Response: We apologize for the lack of clarity in the description of incentive structure. We have added a specific description of the participation payment (line 278): “The final payment was calculated based on the sum of all accepted offers.”.

4. The rationale for using a threshold of 5 to define high SES is unclear. Other criteria such as the mean or median SES score could also be considered. The authors should report the median SES value and clarify whether the results are robust to alternative thresholds (e.g., using the mean or median). Would the main findings hold under a different SES cutoff?

Response: We really appreciate your identification of this issue in our study. Following your constructive suggestion, we calculated the mean and median of SES and used them as the rationale for determining the threshold. The mean is 5.07, median is 5, and SD is 2.07. As the mean is 5.07 and the median equals 5, adopting either as the threshold would result in the same group assignment as the original. And thus, the original findings would remain unchanged.

We have made revisions to the manuscript to clarify the rationale for using 5 as the cut-off point, supported by both the mean and the median (lines 251–253): “To determine the threshold for the high and low groups, we calculated the mean and median of subjective SES, which were 5.07 and 5, respectively. Therefore, we set 5 as the cut-off point.”.

Minor Comments:

1. Does the mention of statistical significance in line 252 refer to the 10% level? Please clarify.

Response: We sincerely appreciate your identification of this issue and apologize for the lack of clarity in the description. A 5% significance level was used in this study, and we have revised the wording to provide greater clarity (lines 322–324): “it was statistically significant as the 95% bootstrap confidence interval did not include zero (β = -0.0016, 95% CI = [-0.0032, -0.0004]),”.

2. Reference 30 appears to be missing from the list.

Response: We apologize for the missing reference. We have completed it.

3. The reference list should be ordered numerically.

Response: Thank you for pointing out this issue, we have revised the reference list.

Reviewer 2 comments:

1. I was expecting that depression would be a predictor of UG rejection rates, although in your introduction you wrote that there is mixed evidence of this relationship. This result has some implications for the hypothesis tested, but it is not directly stated in the results or discussed in context of the previous literature. I would like you to add a bit more in the results and discussion regarding the lack of this effect and why that might be.

Response: We express our gratitude to you for astutely acknowledging this observation. We have revised the manuscript to emphasize this point in both the Results and Discussion sections.

In Results (lines 324–328): “And the direct effect of depression on the rejection rate of unfair offers was not significant (β = 0.0007, 95% CI = [-0.0031, 0.0044]), suggesting the presence of full mediation. This result indicates that depression may influence rejection behavior fully through the fairness perception of unfair offers.”.

In Discussion (lines 387–396): “While the indirect effect was significant, the direct effect of depression on the rejection rate of unfair offers did not reach statistical significance. This result suggests that the direct link between depression and the rejection of unfair offers is relatively limited. As previous research has found, although depression can induce stronger negative emotions toward unfair offers, it did not directly lead to more rejection tendencies [13]. These findings further elucidate the cognitive-behavioral mechanisms underlying the relationship between depression and the rejection to unfair offers, suggesting that such rejection is not merely driven by negative emotional responses but involves more nuanced cognitive evaluations [40].”.

2. Is there any reason to believe that depression would affect subjective SES? Given that we expect some distortion about the perceptions of the world in depression, could that alter the subjective experience of SES? Could you provide more information about the relationship between the two?

Response: We greatly appreciate your insightful suggestion. These two factors have indeed received a lot of attention, and there is a lot of evidence that they are negatively correlated.

However, the present study found no significant association between depressive symptoms and subjective SES (r = 0.05, p > 0.05). This may reveal the particularity of subjective SES in university students. We have further emphasized and explained this in the Discussion section (lines 483–495): “In addition, this study yielded a noteworthy finding: no significant association was observed between depression and subjective SES (r = 0.05, p > 0.05), which contrasts with previous literature that has typically reported a negative correlation between depressive symptoms and subjective SES [30,31]. This may be because the status of university students' subjective SES in their psychological structure is peculiar, resulting in less correlation with emotional status. Depression among university students is often influenced by short-term environmental factors such as academic pressure and……”.

3. I’d like if the model could include the standardized beta weights in the figure 1 to make it easier to interpret. While it’s not necessary, you may also want to consider other tools to build the model such as powerpoint or google slides to make that easier.

Response: Thank you for this helpful suggestion. We have revised fig 1 accordingly. In addition to standardized beta weights, we also added conditional effects to more clearly present the main findings of this study.

4. Overall, the article could use some additional editing for grammar and clarity. There are several incomplete sentences (ex. “And fairness perception of unfair offers had a negative effect on rejection rate on unfair offers (β = -0.051, p < 0.01).”) that make it a bit harder to read. The writing in the introduction and discussion could also be more concise to get the point across as well. Right now, I had to read it several times to make sure I understood the argument being made. I believe you are arguing that the inconsistent relationship between depression and acceptance of unfair offers is due to differences in perceptions of fairness affected by the interaction of life experiences and cognitive distortions in depression, rather than solely depression. However, there were some other parts of the discussion that I had trouble fitting into the argument, such as the influence of medication and defense mechanisms. Please review these sections to ensure that they are concise and clear about the interpretation of your results.

Response: We sincerely appreciate your identification of this issue and apologize for the lack of clarity in the description. This kind of sentence is due to confusion of prepositions (ex. “And fairness perception of unfair offers had a negative effect on rejection rate on unfair offers (β = -0.051, p < 0.01).”). We have now changed all the ‘rejection rate on unfair offers’ to ‘rejection rate of unfair offers’ in the manuscript for better clarity. We also made extensive modifications to sentence construction and lexical selection to facilitate better understanding.

To improve the clarity and rigor of our arguments, we have revised and expanded the Introduction section accordingly:

i. We have introduced the past research background to explain the impact of depression on unfair rejection from an emotional perspective, and further highlighted the limitations of previous perspectives (lines 86–98): “Considering the key role of negative emotions in rejection to unfairness, the inability to properly process negative emotions caused by unfair treatment may be the cause of depression-related decision-making biases in the UG [5]. Moreover, cognitive models of depression propose a relationship between distorted emotion processing and decision-making, indicating that depressive…...”.

ii. And we have also explained why depression is also related to fairness perception (lines 110–113): “It is worth noting that engaging in altruistic punishment relies not only on emotional motivation, but also on perception of fairness [18]. Depression, which is also associated with cognitive distortions [19], may alter the processing of unfair offers, potentially reducing the likelihood of engaging in punishment behavior.”.

iii. We then added the necessity to introduce subjective SES and its potential mechanisms of interaction with depression (lines 161–173): “Overall, an individual's SES may also influence the processing of unfair offers in the UG, thereby affecting decision-making. Given the profound and prominent nature of this personal attribute in socioeconomic interaction, it should not be overlooked in the UG study. Therefore, SES and depressive symptoms are likely to synergistically shape fairness perceptions, which in turn affect UG decision-making. It is important to figure out how SES intervenes……”.

We have also revised the Discussion section:

i. Prompted by your comment, we recognized that the discussion of medication is not suitable for inclusion here and have removed it. Instead, we proposed that the severity of depression may account for the discrepancy across studies (lines 419–432): “However, studies involving clinically diagnosed MDD samples have found opposite findings. As reported by Wang et al. (2014), Jin et al. (2022) and Scheele et al. (2013), individuals with MDD exhibited higher rejection rates in response to unfair offers [15,16,20]. This implies that the severity of depression may exhibit a non-linear relationship with the tendency to reject unfair offers…...”.

ii. We have explained in more detail why this maladaptive cognitive reappraisal is an emotion regulation defense mechanism (lines 408–412): “Specifically, individuals with depression may actively rationalize unfair offers as more acceptable in order to reduce their emotional conflict and avoid internal distress. They may employ cognitive emotion regulation strategies, such as reappraisal, to mitigate their emotional responses by reframing the situation as “It’s not that unfair after all.”.” And (lines 464–473): “Taken together, we propose that cognitive reappraisal may be a key factor underlying the distortion of rejection behavior toward unfair offers in individuals with subclinical depression. This cognitive reappraisal primarily functions by reframing unfairness as fair, in order to alleviate the negative emotions and internal conflict elicited by unfair treatment. Although it functions as a defense mechanism as it quickly relieves discomfort [45], it does have an impact……”.

5. There was a specific sentence that I found difficult to parse: “Depression is a defense mechanism for fairness perception”. I think this might be reversed from your meaning, although I’m not entirely sure. From the other points you’ve made, I would have thought that cognitive distortions to view the world as more fair would be the defense mechanism for depression. Could you please clarify?

Response: We sincerely apologize for the confusion caused by our unclear phrasing.

Yes, your idea is very accurate and in line with the point we want to make. We have revised the relevant section accordingly for better understanding (lines 408–412): “Specifically, individuals with depression may actively rationalize unfair offers as more acceptable in order to reduce their emotional conflict and avoid internal distress. They may employ cognitive emotion regulation strategies, such as reappraisal, to mitigate their emotional responses by reframing the situation as ‘It’s not that unfair after all.’.” And (lines 464–473): “Taken together, we propose that cognitive reappraisal may be a key factor underlying the distortion of rejection behavior toward unfair offers in individuals with subclinical depression. This cognitive reappraisal primarily functions by reframing unfairness as fair, in order to alleviate the negative emotions and internal conflict elicited by unfair treatment. Although it functions as a defense mechanism as it quickly relieves discomfort [45], it does have an impact……”.

6. Please provide a sensitivity analysis for the moderated mediation based on your sample size.

Response: Thank you very much for your valuable recommendation, which provides important methodological support and assurance for the study employing moderated mediation model.

We incorporated an R²-based sensitivity analysis (Cinelli & Hazlett, 2020) into the data analysis and results sections. This analysis assesses the robustness of key paths in our moderated mediation model against potential omitted variable bias. The results provide additional support for the stability of our findings, showing that unobserved confounders would need to explain a substantial proportion of the residual variance in both the predictor and the outcome to invalidate the observed effects. Relevant details have been added to the revised manuscript.

In Data analysis section (lines 289–300): “To evaluate the robustness of the estimated effects to potential omitted variable bias, a post-hoc sensitivity analysis based on the approach of Cinelli and Hazlett (2020) was conducted using the R package sensemakr [36]. This analysis quantifies how strongly an unmeasured confounder would need to be associated with both the treatment and outcome to invalidate the causal conclusions. Partial R² and robustness value (RV) were reported to assess the vulnerability of the effect, higher RV indicate greater robustness to omitted variable bias……”.

In Results section (line

---

## [Editor Report · Decision Letter 1]

7 Aug 2025

Subjective socioeconomic status moderates depression's impact on fairness perception in the ultimatum game: a moderated mediation model

PONE-D-25-24754R1

Dear Dr. Subhi,

We’re pleased to inform you that your manuscript has been judged scientifically suitable for publication and will be formally accepted for publication once it meets all outstanding technical requirements.

Kind regards,

Caleb Cox

Academic Editor

PLOS ONE
---

## [Editor Report · Acceptance letter]

PONE-D-25-24754R1

PLOS ONE

Dear Dr. Subhi,

I'm pleased to inform you that your manuscript has been deemed suitable for publication in PLOS ONE. Congratulations! Your manuscript is now being handed over to our production team.

Kind regards,

on behalf of

Dr. Caleb Cox

Academic Editor

PLOS ONE